# Spatially Guided and Single Cell Tools to Map the Microenvironment in Cutaneous T-Cell Lymphoma

**DOI:** 10.3390/cancers15082362

**Published:** 2023-04-18

**Authors:** Eirini Kalliara, Emma Belfrage, Urban Gullberg, Kristina Drott, Sara Ek

**Affiliations:** 1Department of Immunotechnology, Faculty of Engineering (LTH), University of Lund, 223 63 Lund, Sweden; 2Department of Dermatology and Venereology, Skane University Hospital (SUS), 205 02 Lund, Sweden; 3Department of Hematology and Transfusion Medicine, Skane University Hospital (SUS), 205 02 Lund, Sweden

**Keywords:** biomarker discovery, cutaneous T-cell lymphoma, mycosis fungoides, patient prognostication, personalized medicine, spatially resolved transcriptomics

## Abstract

**Simple Summary:**

While most patients with cutaneous T-cell lymphoma (CTCL) may be diagnosed with early-stage disease, approximately 25–30% of those patients will unexpectedly progress to the advanced stage with an unforeseeable course of progression and response to treatment. Therefore, it is of pivotal importance to decipher the exact biological events governing disease aggressiveness to identify early those patients who will progress, and to design personalized treatment strategies for them. We propose that the way forward should entail a combination of advanced spatially resolving and tissue disruptive single-cell transcriptomics tools to enable deep phenotypic and molecular profiling of the benign immune and malignant T-cell populations in the two different disease compartments and thus acquire a global view of the inter-patient and intra-tumor heterogeneity. This will promote the development of novel molecular biomarkers for improved prognostication and personalized treatment to improve the survival outcomes and quality of life of chronic cancer patients with MF and SS.

**Abstract:**

Mycosis fungoides (MF) and Sézary syndrome (SS) are two closely related clinical variants of cutaneous T-cell lymphomas (CTCL). Previously demonstrated large patient-to-patient and intra-patient disease heterogeneity underpins the importance of personalized medicine in CTCL. Advanced stages of CTCL are characterized by dismal prognosis, and the early identification of patients who will progress remains a clinical unmet need. While the exact molecular events underlying disease progression are poorly resolved, the tumor microenvironment (TME) has emerged as an important driver. In particular, the Th1-to-Th2 shift in the immune response is now commonly identified across advanced-stage CTCL patients. Herein, we summarize the role of the TME in CTCL evolution and the latest studies in deciphering inter- and intra-patient heterogeneity. We introduce spatially resolved omics as a promising technology to advance immune-oncology efforts in CTCL. We propose the combined implementation of spatially guided and single-cell omics technologies in paired skin and blood samples. Such an approach will mediate in-depth profiling of phenotypic and molecular changes in reactive immune subpopulations and malignant T cells preceding the Th1-to-Th2 shift and reveal mechanisms underlying disease progression from skin-limited to systemic disease that collectively will lead to the discovery of novel biomarkers to improve patient prognostication and the design of personalized treatment strategies.

## 1. Introduction

### 1.1. Mycosis Fungoides and Sézary Syndrome

Primary cutaneous T-cell lymphomas (CTCL) are a group of extranodal non-Hodgkin lymphomas originating from the uncontrolled proliferation of skin-resident or skin-homing T lymphocytes [1,2]. CTCL accounts for approximately 65–75% of all primary cutaneous lymphomas (CL) and constitutes a group of malignancies with heterogeneous clinical, histopathological, and immunophenotypic characteristics [1,2,3,4]. The incidence rate of CTCL has increased in the last decade partly due to better diagnosis of the disease and it may range from six to nine per million people when comparing African Americans and Caucasians, respectively [5,6,7]. Classical subtypes of CTCL include mycosis fungoides (MF) and Sézary syndrome (SS), which manifest male predominance with an approximate 2:1 male to female ratio and mostly affect elderly people with a median age at diagnosis of 50–60 years [8,9,10]. While MF is the most frequent subtype and comprises 60% of CTCL, SS is a rarer disease that accounts for 5% of CTCL cases [10,11]. The remaining subtypes consist of cutaneous CD30+ lymphoproliferative disorders (LPDs) involving primary cutaneous anaplastic large cell lymphoma and lymphomatoid papulosis that account for approximately 25% of CTCL, making CD30+ LPDs the second most frequent subtype. Even more variants exist, and each represents less than 2% of CTCL cases [12,13,14].

MF and SS are regarded clinically and pathologically as two closely related diseases that arise from mature skin-resident or skin-homing CD4+ T lymphocytes [15]. Although initially considered continuums or different stages of the same disease, immunophenotypic analyses have demonstrated that malignant T cells in MF and SS originate from distinct T cell populations. While MF is characterized by CD4+ T cells mainly localizing in the skin and presenting a phenotype of effector memory T cell (Tem), in SS, clonal malignant T cells circulating in peripheral blood manifest a phenotype of central memory T cell (Tcm) [16].

Most commonly, MF has an indolent clinical course with chronic manifestation due to its natural history of slow progression [8,11]. In the early stages of MF (stage IA), skin involvement is presented with limited patches or plaques (designated as T1) that involve less than 10% of affected body surface area (BSA) and patients show a favorable prognosis with a 5-year overall survival (OS) of 94% [8,17]. Similarly, patients with generalized patches and plaques (designated as T2) that involve more than 10% of BSA (stage IB) still present a good prognosis with a 5-year OS of 84% [8,17]. However, in about 25–30% of early-stage patients, the disease may progress into infiltrating plaques, ulcerated tumors (designated as T3), and blood/systemic involvement, and the 5-year OS is reduced to 18% (stage IVB) [8,11]. As briefly mentioned above, CD30+ LPDs account for the second most common CTCL subtype after MF, which histologically presents large CD30+ tumor cells and is characterized by an indolent course and a favorable prognosis [5]. While large cell transformation (LCT) is defined by CD30− or CD30+ large cells, with a cell size fourfold larger than a small lymphocyte, that exceeds 25% of the tumor infiltrate, this histologic feature is identified in only 25–50% of MF patients with advanced stage [18,19,20]. In particular, advanced-stage MF patients with LCT demonstrate a 5-year OS of less than 20% [19], and the identification of LCT soon after disease diagnosis or at an advanced stage has been correlated to a poor disease outcome [21].

SS is more aggressive than MF with a dismal 5-year OS of 30% and high relapse rates [3,22]. It is defined as a leukemic variant of CTCL and characterized by the triad of erythroderma, lymphadenopathy, and the presence of neoplastic T cells of atypical morphology (designated as Sézary cells), which demonstrate the same clonal T-cell receptor (TCR) gene re-arrangement in the skin, peripheral blood and lymph nodes [10].

Although significant progress has been made in understanding CTCL pathogenesis, the exact molecular mechanisms underlying disease development have not been fully elucidated yet. Prior extensive research using conventional genomic approaches as well as advanced whole genome-based sequencing technologies have not identified homogeneously recurrent cytogenetic alterations accountable for the etiology of the disease, while there is no strong evidence for a genetic inheritance to be the dominant pathogenetic factor of CTCL development [23,24,25]. However, genes involved in cellular processes such as DNA damage response, epigenetic regulation, T-cell receptor (TCR), Nuclear factor Kappa B (NF-kB) and Janus kinase (JAK)/signal transducers and activators of transcription (STAT) signaling pathways have been frequently identified as deregulated in MF and SS [14,26,27,28,29].

### 1.2. Patient Risk Stratification: A Clinical Unmet Need in CTCL

Staging criteria are shared between MF and SS and include evaluation of the disease present in the skin (T), lymph nodes (N), viscera (M), and blood (B) [30]. The TNMB classification system essentially assigns patients with early (IA-IIA) or late-stage disease (IIB-IVB). Prognostic indices to stratify patients according to the risk of disease progression have been previously developed primarily through means of retrospective studies [8,9,30]. In particular, the development of the cutaneous lymphoma international prognostic index (CLIPi) predicts survival and risk of progression in patients with early and late-stage disease [9]. However, most of these risk predictive factors are associated with clinico-pathologic characteristics of the disease and rely on limited-size cohorts from single-center studies. Consequently, there is still a clinical unmet need to develop molecular biomarkers associated with disease progression to be utilized in reliable pre-selection of patients who will transition from early to advanced-stage disease and enable personalized treatment decision making to improve disease control and patient survival outcomes.

## 2. The Tumor Microenvironment (TME) and Immune Response in CTCL

While the exact molecular events underlying CTCL progression remain poorly understood, prior research has reported that the tumor microenvironment (TME) plays a pivotal role [23,31,32,33]. During early stages, CD8+ resident memory T cells (T_RM_) harboring cytotoxic capacity and a skewed T helper (Th) 1 immune response are predominantly observed in the skin microenvironment, while the number of neoplastic T cells remains relatively low [34,35] (Figure 1b). The production of CXCL9, CXCL10, and CXCL11 chemokines by keratinocytes and dermal fibroblasts potently promotes Th1-type cell recruitment, the latter exerting a protective immune response to maintain control over tumor cell growth and expansion. However, as the disease progresses, activated keratinocytes play a central role in fostering tumor cell growth and survival via recruiting Th2-type cells in the skin through the production of Th2-type chemoattracting molecules including CCL17, CCL22, and CCL27 that lead to a dampening of the Th1-type anti-tumor immune response [36]. At the same time, periostin production by cancer-associated fibroblasts (CAFs) and subsequent induction of thymic stromal lymphopoietin (TSLP) synthesis by keratinocytes create a positive feedback loop for continuous immunomodulation of the skin microenvironment (Figure 1c). Additionally, dendritic cell (DC) activation potently propagates Th2-type inflammation and mediates cell–cell antigen activation of T cells as previously reviewed elsewhere [23,36,37,38,39]. Essentially, during disease progression, a reduction in the number of cytotoxic CD8+ T cells and an increase in the atypical CD4+ T cell infiltrate takes place accompanied by a shift from an anti-tumor Th1 to a tumor-promoting Th2-type immune response [38] (Figure 1a–c).

Patients with benign conditions including psoriasis and early-stage CTCL patients show high expression of Th1-type cytokines, including IL-2 and IFN-γ, while late-stage patients without blood involvement demonstrate high expression levels of Th2-type cytokines, among which are IL-5, IL-10, IL-13, and IL-17 [40]. Interestingly, late-stage CTCL patients with blood involvement present the most notably suppressed production in most of the above cytokines, with the exception of IL-5 [40]. In line with the late-stage patients, aggressive types of CTCL, including SS, are governed by a predominant Th2-type immune response reflected by aberrant secretion of the Th2-type cytokine, IL-4, and decreased production of Th1-associated cytokines, including IL-12 and IFN-γ [41]. Along these lines, Geskin and colleagues previously documented the importance of the synergistic effect of the Th2-type cytokines IL-13 and IL-4 in promoting tumor cell growth and proliferation in CTCL [42]. More specifically, it was reported that IL-13 is produced by malignant lymphoma cells and acts as an autocrine factor regulating tumor cell proliferation through IL-13R signaling in both CTCL skin lesions and SCs in blood [42]. Together with previous reports, there is compelling evidence that disease progression is correlated with a Th1 to Th2 cytokine shift, a progressive defect in cytokine expression, and an overall immunosuppression towards advanced stages [41,43,44,45].

Malignant T cells have been shown to exhibit an exhausted phenotype, resembling regulatory T cells (Treg), in patients with late-stage disease [40]. The latter are characterized by low expression of cytokines including IL-2, IL-4, and IFN-γ upon TCR stimulation, similar to findings for malignant cells in advanced stages with blood involvement [40]. However, expression of Treg-like markers, as the key transcription factor FOXP3, is low or absent in the clonal malignant T-cell population [46,47,48,49]. Although the expression of FOXP3 in malignant cells in SS and MF is still under controversy, increased infiltration of FOXP3+ Tregs in tumors might be linked to the early-disease stage and improved patient survival [46]. On the other hand, Gaydosik and colleagues showed, using single-cell RNA sequencing (scRNA seq) in skin biopsies, that CD8+ tumor-infiltrating lymphocytes (TILs) expressed genes encoding for co-inhibitory receptors including PD1, CTLA4, TIME3, LAG3, and TIGIT at variable levels across five advanced-stage CTCL patients [50]. Additionally, reactive CD3+CD4+ T cells co-expressed *FOXP3*, *IL2RA*, *PD1*, *CTLA4*, *LAG3*, and *TIGIT*, which implied the presence of benign TILs of Treg phenotype. These findings supported that the reactive lymphocytic infiltrate of the skin microenvironment demonstrates an immune-suppressive profile, particularly in advanced stages [50]. However, it is still to be determined if malignant T cells originate from Tregs or gain Treg-like properties during disease progression according to recent single-cell transcriptomic studies [51].

A recent single-cell study by Du et al. additionally reported the enrichment of T/NK and myeloid cells as a common feature in the skin ecosystem of advanced-stage CTCL patients [52]. More specifically, enhanced crosstalk between malignant CTCL populations and CCL13+ mono/macrophages as well as LAMP3+ conventional DCs (cDCs) were strongly identified across advanced-stage patients. By disentangling intercellular communications in the CTCL skin microenvironment, the inhibitory interactions of CD47-SIRPA, MIF-CD74, and CCR1-CCL18 interactions were found to take place between malignant T cells and mono/macrophages [52]. Additionally, a potent interaction between the most highly expressed gene in the malignant T cell compartment, that of S100A9, and toll-like receptor 4 (TLR4) notably enriched in the CCL13+ mono/macrophage population was detected across CTCL samples, with this axis previously reported to have a tumor-promoting and immunosuppressive role in other cancer types [53,54]. Taking into consideration these findings, the myeloid compartment also exhibits a potent role in shaping an immunosuppressive environment during CTCL progression, while a deeper understanding of the mechanistic pathways involved in the crosstalk between malignant T cells and benign immune cells holds promise to lead to a better understanding of CTCL development and progression and to unravel novel immunotherapeutic directions in advanced CTCL.

**Figure 1 cancers-15-02362-f001:**
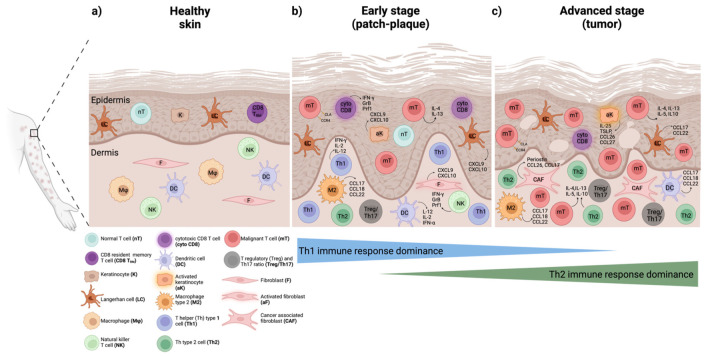
Phenotypic and functional changes in the benign immune cell subsets in the skin microenvironment during MF and SS progression. (**a**) Healthy skin normally containing very few resident immune cell populations in the epidermis and dermis. (**b**) Early stages (patch-plaque stage) characterized by few malignant T cells and a predominant anti-tumor Th1 immune response with cytotoxic CD8 T cells (cyto CD8), Th1 cells, natural killer (NK) cells, and dendritic cells (DCs) playing an important role in restraining malignant T cell proliferation and survival. (**c**) Advanced stages (tumor stage) featured by an increased infiltration of benign immune cells and a dominant Th2 immune response. Here, Th2 cells, M2 type macrophages (M2), cancer-associated fibroblasts (CAFs), and DCs in the dermis, as well as activated keratinocytes and Langerhans (LCs) in the epidermis, contribute to malignant T cell expansion and disease aggressiveness. Changes in the frequency and abundance of regulatory T cells (Tregs) and Th17 cells shown to accompany disease progression. Production of pro-inflammatory, anti-tumor, and tumor-promoting effector molecules demonstrated for each cell type as previously reviewed elsewhere [23,37,55]. The figure was generated using BioRender software.

## 3. Inter-Patient and Intra-Tumor Heterogeneity in MF and SS

### 3.1. Immunophenotypic Heterogeneity

MF and SS exhibit substantial heterogeneity as observed both in the immunophenotype and transcriptional expression profile of the malignant T-cell population between the different disease compartments, over time and across patients [50,56,57]. Roelens and colleagues showed that the phenotype of blood circulating Sézary cells (SCs) largely varied in the expression levels of naïve and central memory markers, while one patient also presented a shift towards a phenotype of stem-cell memory (Tscm), during a six-month follow-up [56]. Skin versus blood-derived SCs were shown to markedly differ in their maturation status, with skin-derived SCs presenting a more differentiated phenotype than the clones in blood, based on the progressive T-cell differentiation model [58]. Additionally, differences in the cytokine and chemokine-receptor expression profile were found in the two anatomically distinct malignant T-cell subsets, a phenomenon previously shown to affect the local efficacy of applied treatment [59]. Consequently, in-depth characterization of the relationship between skin and blood compartments may encompass potential implications in the discovery of new therapeutic targets in CTCL.

Of specific interest, Immunophenotypic switch (IS), a rarely reported phenomenon in CTCL, has been previously identified in up to twenty-five CTCL patients [60]. IS is defined as a change in the immunophenotype of tumor cells while maintaining their genetic expression profile. The most common IS reported particularly in MF patients comprises the change from a CD4+ phenotype to a cytotoxic CD8+ phenotype, especially after treatment [61,62,63,64,65,66,67]. IS has been more recently documented in a series of three cases of CTCL patients presenting disease relapse post-treatment [60]. In many patients, IS has been followed by patients succumbing to their disease, thus highlighting that IS could potentially carry an unprecedented prognostic value and predict a poorer disease outcome.

### 3.2. Genomic Heterogeneity and Subclonal Evolution

Cristofoletti and colleagues presented that skin versus blood-derived SCs differ in their proliferation capacity, with the former exhibiting a higher proliferation index (PI) than blood-circulating malignant cells [57]. Interestingly, the increase of the PI in skin was related to an increase in the tumor burden in blood. Elevated PI in the skin compartment was also accompanied by activated mTOR, a member of the PI3K/AKT/mTOR signaling pathway and central regulator of cell survival and proliferation. Increased mTOR activation was associated with gene copy number (CN) changes in other members of the mTOR pathway [57]. While CN loss of PTEN, an inhibitor of mTOR signaling, and loss of *PDCD4* were identified, CN gain instead was detected for P70S6K in SS patients. Most notably, patients could be categorized into groups with different survival outcomes depending on their CN variation pattern. Patients without the previously defined genetic aberrations presented a median OS of 73 compared to 59 months for patients with one or two of these genetic aberrations. Strikingly, patients harboring genetic aberrations within all three genes presented the worst clinical outcome with a median OS as short as 25.7 months [57]. These findings highlighted that deep genomic profiling of the two disease compartments could reveal biological information with previously unknown prognostic value in CTCL patients.

The profound genomic heterogeneity within the malignant T-cell population is evident and of potential clinical relevance to predict drug resistance in advanced-stage disease [68,69]. Iyer and colleagues identified that as the disease progresses from early-plaque (stage I) to advanced tumor-stage (stage ≥ IIB) in MF skin lesions, the mutational burden of malignant cells is proportionally increased [68]. Phylogenetic analysis and evaluation of the mutational burden distribution revealed that the increase in the genomic variability correlated with an increase in the malignant subclones present within each MF patient. By determining the phylogenetic relationships between the genetically distinct subclones present in different skin lesions in the same patient, the authors proposed a model of divergent evolution for the emergence of multiple neoplastic subclones during disease progression. Essentially, in this model, each lesion is proposed to evolve in relative isolation from other lesions during sequential rounds of re-circulation and re-seeding of malignant T-cell clones and accompanied by further accumulation of genomic mutations leading to subclonal evolution. Herrera and colleagues also found multiple molecular subclones in the skin and blood of late-stage CTCL patients [69]. By determining tissue-dependent signatures, more active and proliferative malignant subclones were identified in skin compared to blood, leading the authors to suggest a model in which the skin microenvironment, where proinflammatory signaling and chronic activation of T cells takes place due to constant exposure to environmental cues and pathogens, supports rapid malignant cell expansion during disease progression [69].

Using whole-exome sequencing (WES), Song et al. also revealed significantly higher tumor mutation burden in a subgroup of patients with transformed CTCL (tCTCL) when compared to SS, the former a CTCL subtype that is featured by the histologic feature of LCT in skin and presents aggressive clinical behavior [70]. High intratumoral heterogeneity was also demonstrated in patients with tCTCL, in which multiple genetically distinct subclones of malignant T cells were identified in the skin of the same patient despite confirmed TCR monoclonality [70]. Interestingly, patients with an increased number of genetically distinct malignant T-cell subclones demonstrated higher tumor burden and worse clinical outcomes, whereas identification of a dominant ultraviolet (UV) mutational signature was associated with favorable prognosis in patients with tCTCL. Furthermore, a few putative gene drivers and recurrently mutated pathways were also identified including Hippo and Notch pathways, which are specifically known to contribute to stemness and metastasis and could be of interest to further explore as future therapeutic interventions in more aggressive CTCL subtypes [70].

### 3.3. Transcriptional Heterogeneity

Similarly in the mRNA compartment, there is a high degree of heterogeneity as shown by Borcherding et al. through the application of scRNA and TCR sequencing in sorted SCs [71]. Here, SCs were transcriptionally categorized into five distinct clusters. Interestingly, machine-learning dimensional reduction showed that individual SCs hold the capacity to differentiate into two different cellular fates. Differential gene expression analysis performing pseudo-time construction indicated that SCs may initially express a FOXP3+IL7RlowTIGIT+ phenotype, with *FOXP3* expression decreasing towards terminal cellular states [71]. This data suggested that although clonally expanded, SCs potentially differentiate transcriptionally over time. In an external cohort of patients, *FOXP3* was the key predictive gene in the classifier along with *TGFB1*, *CD7*, *PTPN6*, and *SUZ12*. Whereas the highest expression levels of *FOXP3* and *PTPN6* were associated with early-stage disease, increased expression levels of *TGFB1*, *CD7,* and *SUZ12* were associated with late-stage disease [71]. Implementation of the aforementioned markers succeeded in correctly classifying 79.6% of patients into early and late-stage disease. However, larger datasets are needed to further validate the sensitivity and prognostic power of the biomarkers discovered. Considering that the same authors more recently revealed that the transcriptional state of malignant T cells shifts towards a Treg-like phenotype post-treatment in one CTCL patient [72], there is additional evidence for the evolutionary capacity of the malignant T-cell population.

Using paired scRNA and TCR sequencing, inter-patient heterogeneity at the transcriptional level was demonstrated by Ren et al. in the blood-circulating malignant T-cell population, whereas substantial similarities were shared among the normal T cell counterparts across 11 patients [73]. Application of the potential of heat diffusion for affinity-based trajectory embedding (PHATE) analysis showed the presence of a pre-cancerous population featured by an intermediate phenotype at the transcriptional and genetic mutational level when compared to normal T cells and clonal CTCL cells. This data suggested a possible transitional population that may be involved in CTCL development, which warrants further investigation. In agreement with previous reports, within the malignant T-cell population among the most enriched pathways in the malignant T cells were the TCR and JAK/STAT signaling pathways. Several exhaustion markers, including *TOX*, *TIGIT*, *LAG3*, *CTLA4*, and *PDCD1* were significantly increased in the clonal malignant T-cell population, prompting the authors to propose that malignant CTCL cells might acquire an exhausted phenotype due to chronic antigenic stimulation during CTCL development [73]. For example, the *CD82* molecule that encodes a cell surface protein marker found upstream of JAK/STAT and AKT/PI3K pathways was identified and suggested as a novel therapeutic target for CTCL patients. It was shown that JAK-inhibitors play a potential therapeutic role via reducing CD82-dependent malignant T cell proliferation and survival ex vivo [73].

In the study by Litvinov et al., 163 formalin-fixed and paraffin-embedded (FFPE) CTCL biopsies at different clinical stages were investigated by targeted RNA sequencing [74]. Here, among 284 pre-defined genes [75,76], unsupervised clustering analysis failed to discriminate between the three different disease stages both between different patients and within the same patients who progressed over time. Nevertheless, downstream analysis identified *TOX*, *FYB*, and *GTSF1* genes particularly, upregulated in advanced stages compared to intermediate and early-disease stages, while *LTB4* was downregulated in advanced stages of CTCL [74]. Furthermore, differential expression of *CCR4*, *TOX*, *FYB*, *SERPINB3*, and *PPTN6* genes differentiated stage I patients with stable disease compared to stage I patients that later progressed [74], and are interesting candidates for a predictive test.

Similarly, stage-dependent expression signatures were also identified when comparing patch-stage to advanced plaque/tumor-stage lesions in three patients using scRNA and TCR sequencing [51]. While a few commonly upregulated genes, including *CD70* and *GTSF1*, were observed across patients in line with previous reports [74,76], differentially expressed genes including *CXCR4*, *CD69*, *HSPA1A*, *ZFP36*, *IL7R* (CD127), and *TXNIP*, were specifically downregulated in the plaque/tumor-stage compared to early patch-stage lesions, findings which implied possible mechanisms of disease progression with potential prognostic value. Follow-up analysis of the same lesions upon disease progression in one MF patient demonstrated that clonal malignant cells upregulated genes encoding the pro-tumorigenic factor LTB, the NK-associated receptor *KLRC1* (NKG2), and *CD47* molecules [51]. Notably, an increased number of B cells were observed in plaque/tumor-stage compared to patch-stage lesions as well as in late-stage MF, indicating a potential role of B cells during disease exacerbation.

Transcriptional differences in malignant cells derived from biopsies from different anatomical compartments within the same MF patient were more recently reported [77]. In five out of 11 patients, loss of TCR expression was detected in a subset of malignant cells, originally observed in more aggressive lymphomas and supported as a common phenomenon of TCR instability [78,79]. To examine signaling pathways of prognostic value, groups of genes were studied. Genes related to increased T cell signaling and activation, including *HLA-DRB1*, *CD69*, *MYC*, *ITK*, *FYN*, and *CBCL* were associated with a favorable prognosis, whereas genes linked to high T cell activation status, among which were *MCMC7*, *PCNA*, and *BIRC5*, were shown to predict poor patient outcomes [77]. By profiling and comparing the transcriptional activity of malignant cells with their reactive CD4+ or CD8+ T cell counterparts across patients, the authors categorized patients into two groups relative to their molecular expression profile, irrespective of disease subtype. The two different groups presented differences in the expression of molecular markers consistent with either cytotoxic effector memory T cell (T_CyEM_) or T_CM_ phenotype. Malignant cells in patients of T_CyEM_ were featured by the expression of a 19-gene signature characterized by several cytotoxic markers including *GZMA*, *GZMH,* and *NKG7*. On the other hand, malignant cells in patients of T_CM_ phenotype presented expression of a 27-gene signature, including *TOX*, *KIR3DL2*, *CD40LG*, *GTSF1*, *PTTG1*, *GAPDH*, *PPIA,* and *HSPD1*, markers previously described in patients with an advanced CTCL stage [50,71]. Patients harboring a T_CM_ phenotype manifested late-stage disease, and their corresponding expression signature was associated with shorter progression-free survival (PFS). Furthermore, the immune cell composition within the TME of patients was demonstrated to correlate with disease burden. While CD8+ T cells in the T_CyEM_ group exerted a more robust cytotoxic anti-tumor phenotype, CD8+ T cells in the T_CM_ group manifested a highly exhausted phenotype featuring the relatively higher expression of several exhaustion markers as previously described [80], including *PDCD1*, *CTLA4*, *LAG3,* and *TIGIT*, which is in agreement with the advanced stage of the disease identified in this group of patients. Finally, while M2-like macrophage expression signatures were enriched in the T_CyEM_ group, an increased abundance of B cell expression profile was detected in the T_CM_ group of patients [77].

Large inter-patient gene expression heterogeneity in the malignant T-cell population was additionally observed in fresh skin biopsies derived from eight patients with tCTCL in the study by Song et al. [70]. In agreement with the previous report [73], this contrasted the more preserved gene expression profile observed in the normal T cell counterparts across patients. Significant enrichment of genes involved in the oxidative phosphorylation (OXPHOS), MYC, cellular plasticity, and stemness signaling pathways was shown in the malignant T-cell population derived from the transformed tumor (TT) lesions. More specifically, genes including *NDUFB2*, *TWIST1,* and *LGALS3* were identified as drivers in tCTCL development [70]. Additionally, investigation of the tumor immune microenvironment (TIME) in tCTCL demonstrated significant enrichment of B cells in TT lesions in comparison with concurrently existing plaque/patch (PP) lesions in the same patients. These findings suggested a putative tumorigenic role of B cells during CTCL transformation, findings which are in concordance with previously described studies [51,77]. Follow-up receptor-ligand interaction analysis utilizing scRNA seq data combined with in situ immune profiling of matched PP and TT lesions in the same tCTCL patients revealed an important crosstalk between malignant T cells expressing the macrophage migration inhibition factor (*MIF*) and macrophages as well as B cells both expressing *CD74* molecules, a phenomenon more prominently observed in TT compared to PP lesions. Overall, this study proposed putative gene drivers in the development of tCTCL disease, which is considered an aggressive CTCL subtype and additionally highlighted potential molecular mechanisms underlying malignant T cell and benign immune cell crosstalk taking place in the TIME that may have future implications as therapeutic targets in tCTCL [70].

Considering the large genomic and transcriptional heterogeneity governing CTCL among and within the same individual, alongside the emergence of subclonal evolution within the clonal malignant T-cell population, it is becoming apparent that a personalized approach of molecularly profiling the distinct malignant T-cell subpopulations is essential to better design therapeutic strategies, predict a patient’s treatment response and disease outcome. However, a parallel focus should be given to characterizing the immune microenvironment and in particular profiling the cellular and molecular changes in the benign immune subpopulations accompanying disease progression to develop improved molecular biomarkers common across patients for future implementation in clinical routine practice.

## 4. Moving Forward: Deep Immune Profiling of CTCL Heterogeneity by Spatially Resolved Omics Technologies

Recently, the importance of topography and the functional phenotype of immune cells within the CTCL microenvironment was implicated in the prediction of patients’ treatment responses [81]. Skin biopsies from patients with advanced CTCL were evaluated pre- and post-treatment with the anti-PD1 immune checkpoint inhibitor (ICI), pembrolizumab [81]. In this study, simultaneous tissue staining with 55 immunophenotypic markers was conducted using the CODEX platform to discriminate between malignant and reactive CD4+ T cells. Here, differences in the function and spatial organization between tumor-infiltrating CD4+PD1+ T cells, Tregs, and malignant CD4+ T cells were correlated with the treatment response of CTCL patients. The authors developed a simplified score of the spatial relationships between effector, immunosuppressive and malignant cells, namely SpatialScore, via implementing a low-plex immunohistochemistry (mIHC) platform [82], that included labeling with up to eight phenotypic markers. SpatialScore was demonstrated to predict patients’ response outcomes to ICI treatment with an optimal cut-off of sensitivity versus specificity of 0.79 [81]. Collectively, these results suggested that spatial localization of different immune cell types within the TME of CTCL patients could be used as predictive biomarkers to instruct treatment decision making.

### 4.1. Introducing Spatially Resolved Transcriptomics (SRT)

Voted the method of the year in 2020, spatially resolved transcriptomics (SRT) bridges the gap between RNA sequencing approaches lacking spatial information and visualization technologies, including in situ hybridization-based approaches that are limited to the number of analytes examined at a time [83]. Latest advancements comprise in situ capturing/barcoded approaches including 10× Visium, Slide-Seq, High-Definition Spatial Transcriptomics (HDST), and GeoMx Digital Spatial Profiler (DSP), in which visualization of morphological features is performed in tissues of intact integrity and sequencing or quantification of the expression of RNA molecules is subsequently performed ex situ [84,85]. These platforms have revolutionized the depth of spatial resolution and multiplexing capabilities allowing for an unpreceded amount of biological information to be revealed in an unbiased manner. Since the first application in 2016 [86], these technologies have been eagerly adopted in the field of immuno-oncology focusing on how the spatial localization of different immune subpopulations within the TME contributes to disease aggressiveness and shape the response outcomes to anti-cancer immunotherapy [84].

### 4.2. SRT Application in Cancer Disease

SRT offer the unique opportunity not only to visualize and phenotypically characterize variable immune cell phenotypes within the TME, but also to elucidate their functional properties in different anatomical tumor sites by in-depth molecular profiling. In particular, the GeoMX DSP platform allows profiling of the whole transcriptome, as well as proteomic expression analysis in the same or sequential sections of previously hard-to-study FFPE tissues [87]. The application and contribution of DSP in cancer research has been already exhibited in several cancer types among which non-small cell lung cancer (NSCLC) and melanoma are only a few examples [88]. In particular, Zugazagoitia and colleagues reported the identification of a novel panel of 12 biomarkers predictive of clinical response to an anti-PD1 checkpoint blockade in NSCLC patients using DSP technology, independently of conventional clinical prognostic factors [89]. DSP not only allowed the discovery of candidate predictive biomarkers of clinical benefit to anti-PD1 immunotherapy but also directed researchers into additional areas of investigation including the involvement of CD56+ NK and natural killer T (NKT) cells in favoring anti-tumor immune response and therapeutic efficacy within the NSCLC microenvironment [89]. Toki and colleagues utilized the DSP platform to spatially resolve and profile the TME in pre-treatment biopsies of a cohort of anti-PD1 treated melanoma patients [90]. Here, an 11- and 15-immune cell marker expression signature was associated with prolonged PFS and OS, respectively. More importantly, by cellular compartmentalization of the TME, the authors discovered that PD-L1 expression in CD68+ macrophages and not in melanoma cells was strongly associated with a favorable prognosis and treatment response [90]. These findings further highlighted the importance of unraveling the spatial information hidden in the TME, which may relate to the impact of stromal components on patients’ responses to treatment and clinical outcomes.

### 4.3. SRT Application in CTCL

Application of SRT in CTCL will contribute to better deconvolving the heterogeneity in the functional phenotype of different benign subpopulations of activated, exhausted, and inhibitory T cells in the skin microenvironment and characterize their spatial organization and dynamic crosstalk with proximal malignant cells (Figure 2). Such a targeted phenotypic and molecular profiling will help in resolving the biology underpinning the extreme inter- and intra-patient heterogeneity in CTCL. Following patients over time with SRT will contribute to understanding the exact signaling pathways and effector molecules underlying disease progression, including those involved in the Th1-to-Th2 shift in the immune response accompanying disease aggressiveness. Considering the extreme heterogeneity manifested by malignant T cells, focusing the phenotypical and molecular characterization on benign immune populations is a promising approach to identifying common prognostic expression signatures across CTCL patients. Essentially, by answering both the “where” (tissue spatial organization) and “what” (gene expression patterns), SRT will allow for acquiring a holistic view of the disease (Figure 2). Novel findings of such integrated analyses could be easily transferred to the clinics since the above technologies are based on standard histology and next-generation sequencing (NGS) approaches. Hence, currently developed molecular profiling and multicolor fluorescent microscopy platforms that are accessible and readily available on a clinical routine basis could be utilized.

## 5. Conclusions—Future Research Directions

Extreme inter- and intra-patient heterogeneity poses major difficulties in understanding the exact molecular events associated with CTCL progression and hinders the development of reliable prognostic biomarkers. While the emergence of advanced omics technologies has undoubtedly enriched our knowledge regarding disease etiology and development, most of these approaches involve the destruction of tissue architecture, therefore limiting our ability to investigate the role of the TME composition and spatial organization in CTCL progression. We propose here that in future studies the application of SRT in skin biopsies will allow deep phenotypic and molecular characterization of diverse benign immune subpopulations in the skin microenvironment. Considering that blood involvement is associated with disease aggressiveness and progression, tissue-based analyses should be paralleled by single-cell transcriptomics to characterize benign and malignant immune populations in paired blood samples over time. Consequently, spatial profiling of the CTCL skin microenvironment combined with single-cell transcriptomic studies in paired blood samples will provide a global view of the disease heterogeneity at the inter- and intra-patient levels. A schematic illustration of a tentative workflow is shown in Figure 2. This will allow the research community to define novel immune-cell-associated expression signatures that contribute to disease severity and systemic progression that are common across patients, including those preceding and associated with the Th1-to-Th2 shift in the immune response, therefore paving the way towards improved stratification and targeted immunotherapy for CTCL patients.

## Figures and Tables

**Figure 2 cancers-15-02362-f002:**
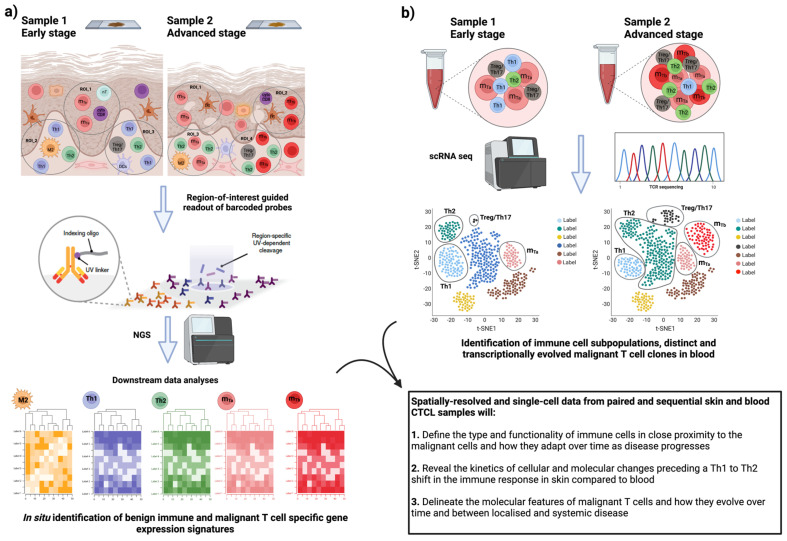
Phenotypic and functional profiling of benign and malignant immune cells in skin and blood CTCL samples with spatially resolved and single-cell transcriptomics. (**a**) Spatially resolved transcriptomics (SRTs) in the skin of MF and SS patients. Shown is an example of a workflow of an SRT technology, in which fluorescently labeled immune cell subpopulations of interest guide gene expression profiling of specific regions of interest (ROIs). Subsequent downstream analysis pipelines shown to assist the identification of gene expression patterns related to specific reactive immune cell subpopulations and molecularly heterogeneous malignant T-cell subpopulations (designated as m_Ta_ and m_Tb_ in pink and red color, respectively) and with respect to their spatial localization in the skin microenvironment in early and late-stage disease. (**b**) Parallel application of single-cell RNA sequencing (scRNA seq) in paired blood samples of CTCL patients. Shown the combined analyses to facilitate the identification of distinct and transcriptionally evolved malignant T-cell clones in relation to simultaneous changes in the frequency, abundance, and molecular profile of the different immune cell populations (Th1, Th2, and Treg/Th17 cells) when comparing early/localized and advanced/systemic disease. Proposed here, the comparison and integration of the findings from sequential skin and blood samples in CTCL patients to allow characterization of the intra-patient heterogeneity and discovery of the unique effector molecules and signaling pathways associated with disease severity and systemic progression over time. The figure was generated using BioRender software.

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
