# Peer review of "Spatially Guided and Single Cell Tools to Map the Microenvironment in Cutaneous T-Cell Lymphoma"

_cancers, 2023, doi:10.3390/cancers15082362_

Round 1

Reviewer 1 Report

Dr. Kalliara and colleagues provide a detailed and well written review of the role of the tumor microenvironment in CTCL, focusing nearly exclusively on MF and SS. 

Unfortunately, several references appear to be missing. 

Furthermore, I would suggest to give a range for the incidence of CTCL, especially as it is geography dependent and has been noted to be rising over the last 2 decades. 

Author Response

  1. We have gone through the review and included references that were missing, see added references 6, 7, 17, 23-29, 42, 52, 70, 73. In particular, lines 89-99 (Ref 23-29) were added to refer to previous studies describing most commonly deregulated pathways in CTCL that have been suggested to be involved in disease pathogenesis. Lines 172-189 (Ref 52) were added to refer to a recent single-cell study focusing on the immune cell compartment and its role in CTCL progression. Lines 279-292 and 386-408 were added to include the recent study by Song et al (Ref 70). Additionally, lines 314-332 were added to include the recent study by Ren et al. (Ref 73).

  1. At line 46-49 the estimated range for the incidence of CTCL has been added.

Reviewer 2 Report

This is a useful summary highlighting approaches to dissecting tumor cell and TME heterogeneity in skin and blood. Lacking is a comparison between patch, plaque and tumor stages of MF in the skin.  There is no discussion of tumor stage CD30+ CTCL and how it compares to large cell transformation of MF.  References to role of IL-13 in growth of MF and SS are missing (Geskin et al, Blood). 

Minor- Line 127- please refer to FoxP3 as a transcription factor, not antigen. Line 264 from not form. Please further explain what is meant by an exhausted T cell phenotype, line 287. Figure 2- mta and mtb cannot be discerned in the different shades of red used. Please use a better format. 

Author Response

  1. We have added information about patch, plaque and tumor stages of MF in the skin, see lines 67-74.

  1. We have added a section on CD30+ CTCL and how it compares to large cell transformation of MF, see line 75-83.

  1. We have added the reference missing from Geskin et al. (Ref 42) (lines 145-150).

  1. Line 158 has been changed to refer to FOXP3 as a transcription factor.

  1. Line 357 – the typo has been corrected.

  1. Line 380-382 – we have added an explanation of exhaustion of T-cells with an additional reference (Ref 80).

  1. Figure 2 has been updated to better separate mta and mtb

Round 2

Reviewer 2 Report

Thank you for addressing recommendations for improvement of the manuscript.